# Tracking Loss: Converting Object Detector to Robust Visual Tracker

## Abstract

In this paper, we find that by designing a novel loss function entitled, "tracking loss", Convolutional Neural Network (CNN) based object detectors can be successfully converted to well-performed visual trackers without any extra computational cost. This property is preferable to visual tracking where annotated video sequences for training are always absent, because rich features learned by detectors from still images could be utilized by dynamic trackers. It also avoids extra machinery such as feature engineering and feature aggregation proposed in previous studies. Tracking loss achieves this property by exploiting the internal structure of feature maps within the detection network and treating different feature points discriminatively. Such structure allows us to simultaneously consider discrimination quality and bounding box accuracy which is found to be crucial to the success. We also propose a network compression method to accelerate tracking speed without performance reduction. That also verifies tracking loss will remain highly effective even if the network is drastically compressed. Furthermore, if we employ a carefully designed tracking loss ensemble, the tracker would be much more robust and accurate. Evaluation results show that our trackers (including the ensemble tracker and two baseline trackers), outperform all state-of-the-art methods on VOT 2016 Challenge in terms of Expected Average Overlap (EAO) and robustness. We will make the code publicly available.

## 1 Introduction

Visual tracking is a fundamental computer vision task, and can be used to predict the trajectory of objects in a video sequence. It is the building block for applications in self-driving vehicles, robotics and automatic surveillance.

The richness of feature representations is crucial to the success of Convolutional Neural Networks (CNNs) in many computer vision tasks, such as image classification and object detection. This property also motivates researchers to adopt CNNs as strong feature extractors in the setting of visual tracking (Wang et al., 2015; Ma et al., 2015; Hong et al., 2015; Nam & Han, 2015; Nam et al., 2016). MDNet (Nam & Han, 2015) is a famous CNN based tracker which achieved tremendous success on VOT 2015 (Kristan et al., 2015). One major drawback is that MDNet is trained with annotated video sequences provided by previous challenges. Therefore, MDNet is lack of generalizability to a diversity of tracking targets. Actually, there is no such dataset with enough labeled video sequences specialized to train trackers.

Previously, a series of trackers (Wang et al., 2015; Ma et al., 2015; Hong et al., 2015) attempted to convert CNN based classifiers trained on large scale image classification datasets (for example ImageNet (Russakovsky et al., 2015)) to tracking. In our opinion, these approaches relied heavily on feature engineering and feature aggregation. That would result in more time and computational cost. Another feasible idea might be to convert pre-trained object detectors to trackers. Region Proposal Network (RPN) (Ren et al., 2015) is a state-of-the-art object detector and achieves tremendous success. It could simultaneously provide strong features for classification and bounding box regression. After a careful exploration, we find the internal structure of RPN is highly relevant and possesses strong potential for discriminative trackers.

Visual trackers usually require to be trained online to learn specific appearances of targets. However, ground truths are quite limited. Data augmentation is widely employed to enlarge training samples.

As to positive training samples, they are a batch of randomly cropped patches, which are subject to a two-dimensional Gaussian distribution, from the entire image around the ground truth. However, this sampling strategy usually contains some background pixels at the border of sampled images. These background pixels are noisy to train a well-performed tracker. With time elapse, accumulative noises will make the tracker get even worse. Due to above problem, if RPN is directly employed in tracking without any modification, different pixels will be treated equally, and the tracker will perform poor. In our opinion, we think pixels in different positions should be treated discriminately. Generally speaking, centric pixels of sampled images are more confident to cover the target object than border ones. Also corresponding feature points of centric pixels are more likely to be positive. In order to realize treating different feature points discriminatively, we explore the top layer feature maps of RPN, design a series of matching strategies, and evaluate them quantitatively and qualitatively. Due to none of the matching strategies is the best, we propose the tracking loss composed of two better performed matching strategies. Such method proves to be effective to take advantage of pros and offset cons of each matching strategy. Tracking loss bridges the gap between object detection and visual tracking in an unconventional loss viewpoint which will not bring extra computation.

RPN is a relative large network which would limit tracking speed. Basing on knowledge distillation theory (Hinton et al., 2015), we also propose a network compression method to trim the RPN. Experiments show that tracking loss would remain highly effective even if the network is drastically compressed.

Furthermore, we adopt a carefully designed tracking loss ensemble which is consist of four types of loss functions. Evaluation results show that tracking loss ensemble could perform much better. Our trackers (including the ensemble tracker and two baseline trackers) outperform all state-of-the-art methods on VOT 2016 Challenge in terms of Expected Average Overlap (EAO) and robustness.

The contributions can be summarized as follows,

- We propose a novel tracking loss which successfully converts a pre-trained object detector RPN to a state-of-the-art visual tracker without extra computational cost. It shed new lights on transferring pre-trained detection network to new tasks where labeled data is very scarce.

- We propose a network compression method to speed up our tracker. Meanwhile, it proves that tracking loss is a robust way to convert detection to tracking and independent of network variations. Furthermore, we implement a tracking loss ensemble with four types of loss functions to further promote tracking performance.

- Our two baseline trackers and the ensemble tracker outperform all state-of-the-art trackers on VOT 2016 (Kristan et al., 2016) in terms of EAO and robustness.

## 2 RELATED WORK

Most visual tracking algorithms can be categorized into two classes, generative or discriminative. In generative approaches (Comaniciu et al., 2003; Zhang et al., 2012; Han et al., 2008), target appearances are trained to represent objects. Trackers search the most matched region as target prediction. Discriminative trackers regard tracking as a classification task. Taking the state-of-the-art MDNet (Nam & Han, 2015) as example, firstly 256 candidate proposals are sampled as network input. Sampling is subject to a two-dimensional Gaussian distribution which are cropped around the target object from the entire image. The mean is the target height and width of previous frame, and standard deviation is (0.15(height + width), 0.15(height + width)). Next, the tracker executes a binary classification to judge whether candidate proposals are the target or background. MDNet was the champion of VOT 2015 Challenge (Kristan et al., 2015). But due to it was trained with labeled video sequences, therefore lack of generalizability to various tracking targets.

If CNN features pre-trained on other tasks can be utilized by visual tracking, that would be quite meaningful. Wang et al. (2015) and Ma et al. (2015) took cross-layer feature selection, but it is hard to manually decide the best lower level features which works well across various scenes and domains. Hong et al. (2015) proposed a target specific saliency map for sequential bayesian filtering by back-propagating relevant features. However, back-propagation is a time consuming operation, thus not proper for speed sensitive visual tracking tasks.

RPN is first adopted in a popular two-stage detector, Faster R-CNN (Ren et al., 2015), which integrates candidate proposal generation and classification into an identical convolutional network. RPN is consist of a backbone network and two following convolutional layers to provide robust features for proposal generation, classification and bounding box regression simultaneously. The backbone network is usually a pre-trained CNN based classification network, such as AlexNet (Krizhevsky et al., 2012), ZF (Zeiler & Fergus, 2014), VGG-16 and VGG-19 (Simonyan & Zisserman, 2014). RPN provides translation invariant anchors. An anchor (Ren et al., 2015) is essentially a 1x1 position in feature maps of a specified layer. Each anchor is conceptually corresponding to a few imaginary areas in input images called as anchor boxes. This property is convenient to map an anchor point to corresponding anchor boxes in the input image, also translate regions to corresponding anchors. In Faster R-CNN (Ren et al., 2015), they assigned 9 kinds of anchor boxes, assembled by 3 types of aspect ratios (1:1, 1:2, 2:1) and 3 scales ($128^2$, $256^2$, $512^2$).

For discriminative trackers, a backbone network is required to extract target independent high-level features for further classification. Furthermore, they should have the ability to learn target specific features for diverse tracking tasks. We find the network structure of RPN is suitable to construct a discriminative tracker. The backbone network in RPN can be utilized to generate target independent features. Two subsequent convolutional layers would be re-initialized to learn target specific features for each tracking video. However, object detection and tracking are two independent tasks. In tracking, the ground truth wouldn't be provided except for the first frame. If converting RPN to tracking directly, loss of bounding box regression will lose efficacy. A feasible idea is to re-design the loss function in RPN to adapt to tracking. To our best knowledge, no previous work attempted to convert a detector into a tracker merely by adjusting loss function.

Larger deep neural networks could enhance model capability, but simultaneously result in more time and computational consuming. RPN is relative large to tracking. Network compression is required to speed up tracking procedure. Moreover, a well-performed loss function should be independent of network variations, just like network compression. Han et al. (2015) proposed a network compression pipeline, including 3 stages: pruning, quantization and huffman encoding. But that pipeline is not proper for CNN based RPN.

Previously, ensemble method has already been employed in tracking. Cao & Xue (2013); Bai et al. (2013); Wang & Yeung (2014) proposed to combine a set of weak classifiers to build a strong tracker. Han et al. (2017) proposed the "BranchOut" to silence a subset of CNN branches during model update to regularize the online ensemble tracker. Apart from network ensemble, loss ensemble can be also a feasible way to strength the tracker. And that is a much lighter method to implement ensemble.

## 3    CONVERTING RPN TO TRACKING

In this section, we will introduce our first trial analysis when directly employing RPN in tracking. We find different feature points should be treated discriminately. After exploring the top layer feature maps of RPN, we design and evaluate four matching strategies according to discrimination in confidence scores. Finally, we propose the tracking loss which is composed of two better performed matching strategies.

### 3.1    FIRST TRIAL ANALYSIS

We observe that if directly employing RPN in tracking without modifications, the tracker would perform terrible. The reasons can be concluded as follows. RPN could provide target independent features. However, tracking targets are various. Target specific appearances require to be learned online for a variety of tracking tasks. But in visual tracking, ground truths are quite limited for online training. Therefore, data augmentation is widely used to enlarge training samples. In MDNet (Nam & Han, 2015), with respect to negative training samples, they are some patches randomly cropped from background. As to positive training samples, they are a batch of cropped patches, which are subject to a two-dimensional Gaussian distribution, from the entire image around the ground truth. If taking this kind of sampling strategy, positive samples will usually contain several background pixels around the border. These background pixels are a kind of noise during online training. With time elapse, accumulative noises will make the tracker get even worse. In our opinion, we think

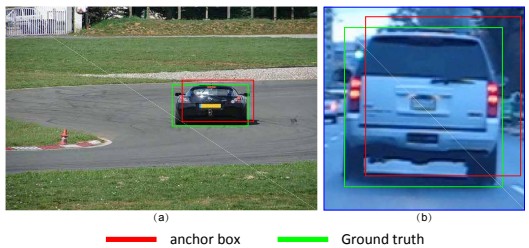

Figure 1: (a) A matched anchor box and the ground truth in object detection. (b) A matched anchor box and the ground truth in visual tracking.

pixels in different positions should be treated discriminately. Due to sampling follows a Gaussian distribution, centric pixels are more confident to cover the target object than border ones for positive samples. Also corresponding CNN feature points of centric pixels are more likely to be positive. In order to treat different feature points discriminately, we explore the top layer feature maps of RPN.

According to the difference in backbone network type, RPN could be different versions. In order to achieve a trade-off between network capability and speed, we select the ZF type (Zeiler & Fergus, 2014) RPN which receives 203x203 sampled RGB images as input.

## 3.2 EXPLORING TOP LAYER FEATURE MAPS OF RPN

The size of the top layer feature maps of ZF type RPN is 14x14, as shown in Figure 2. Each feature point is an anchor. The $receptive\ field$ of an anchor in the top layer of RPN is 171x171. In other words, maximum visual field an anchor can see. Therefore, we set the scale of anchor box and $receptive\ field$ the same. Moreover, we only reserve anchor boxes with aspect ratio 1:1. That significantly reduces the number of anchor boxes from 9 to 1, also a large amount of trainable weights, and accelerates tracking speed a lot.

In RPN, the purpose of anchor boxes is to provide fundamental coordinates in the input image. If the ground truth is given, the similarity between anchor boxes and the ground truth can be calculated to categorize anchor boxes as target object (positive) or background (negative). We use Intersection Over Union (IoU) to define similarity. Figure 1 (a) shows a ground truth (the green rectangle) matched with an anchor box (the red rectangle) in object detection, whose IoU is larger than 0.7. Figure 1 (b) shows the relationship between ground truth and a possible anchor box in tracking.

Although ground truths are not provided in tracking, according to analysis in Section 3.1, centric pixels are more confident to cover the target object than border ones. Also corresponding central feature points of centric pixels are more likely to be positive. With even height and width, the central point of RPN is not unique. Therefore, we regard the union area of central anchor boxes (corresponding to 2x2 anchor points with value 1 shown in Figure 2) as the ground truth. We calculate the IoU of each anchor box with newly defined ground truth as its confidence score, and display it at each point of Figure 2.

We can treat anchors in different positions discriminatively according to Figure 2. A higher confidence score means the corresponding anchor box is much more similar with the ground truth. During training, error back-propagation through these points is more confident. If only considering points with high confidence scores, $blue$ points in Figure 2 have IoUs larger than 0.8. We define the $blue$ area as the first matching strategy, just looking like a symbol +. As to a medium threshold, we choose a square area with least IoU 0.49 as the second matching strategy, including $red$ and $blue$ points in Figure 2. What's more, anchor points contained in another square area, with least IoU 0.24, are defined as the third matching strategy, including $green$, $red$ and $blue$ points in Figure 2. In addition, the full top feature points are defined as the fourth matching strategy, including all $orange$, $green$, $red$ and $blue$ points in Figure 2. During training, only feature points in the matched area would be considered to calculate the total loss in each matching strategy. For the rest mismatched points, they would be ignored. Parameters will only be updated through matched anchor points. Such contraption treats different feature points discriminatively.

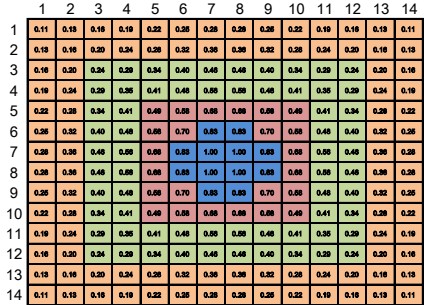

Figure 2: Confidence scores and matching strategies in the top layer feature maps of ZF type RPN. Confidence scores are IoUs of corresponding anchor boxes with defined ground truth. Region in *blue* is defined as the first matching strategy, *red* and *blue* as the second, *green*, *red* and *blue* as the third, full map as the fourth.

### 3.3 LOSS FUNCTION OF RPN IN OBJECT DETECTION

Eq. (1) is the original loss function in RPN (Ren et al., 2015). There are two parts, loss of classification ($L_{cls}$) and loss of bounding box regression ($L_{reg}$). $q_i$ and $t_i$ are the predicted probability and predicted bounding box, and $q_i^*$ and $t_i^*$ are their ground truths. $\lambda$ is a hyper-parameter to balance these two parts. In visual tracking, only the first frame has one labeled bounding box as ground truth. Bounding box regression will lose efficacy in tracking. Therefore, only $L_{cls}$ part is reserved during constructing our RPN based tracker. $L_{cls}$ can be different types of loss function, including Softmax Logistic Loss, Info-gain Loss , Sigmoid Cross-entropy Loss and Hinge Loss.

$$L_{RPN} = \sum_i L_{cls}(q_i, q_i^*) + \lambda \sum_i q_i^* L_{reg}(t_i, t_i^*) \tag{1}$$

### 3.4 MATCHING STRATEGY EVALUATION

**Quantitative Evaluation:** Softmax Logistic Loss is a commonly used loss function among above four types. We implement different matching strategies with it. In order to give a concrete comparison of four matching strategies, we conduct a quantitative evaluation on VOT 2016 benchmark (Kristan et al., 2016). Results are shown in Table 1. There are two types of experiments, unsupervised and baseline. For unsupervised experiment, it is the traditional One Pass Experiment (OPE) without re-initialization. In baseline experiment, trackers would be re-initialized once the predicted bounding box completely drifts off the ground truth. Average Overlap (AO) is an indicator standing for the average IoU for frames across all videos.

Table 1: Matching strategy evaluation results. The first and fourth matching strategies perform better both in baseline and unsupervised experiments. *Red* stands for ranking the first and *blue* the second.

| Experiment (AO) | Baseline | Unsupervised |
|---|---|---|
| First matching strategy | **0.47** | **0.39** |
| Second matching strategy | 0.45 | 0.38 |
| Third matching strategy | 0.41 | 0.36 |
| Fourth matching strategy | **0.47** | **0.43** |
| First + Fourth | **0.52** | **0.46** |

According to Table 1, trackers with the first and fourth matching strategy perform better both in baseline and unsupervised experiments. But their performance still can not reach our expectation.

**Qualitative Evaluation:** If we use a relatively high threshold of IoU during anchor box matching, for example the first matching strategy, less feature points would be used during computing the loss. Total loss reflects more about central points in the feature maps. We observe this kind of tracker is able to successfully follow up most of the objects at the beginning, including very challenging

videos, as shown in Figure 3 (a). However, the bounding box will gradually get larger and larger along with time elapse, as Figure 3 (b). If we use a relative low IoU threshold, for example the fourth matching strategy, although bounding box quality could be improved, the tracker drifts quickly on challenging videos with abrupt motion or deformation, such as *human body* and *hand* shown in Figures 3 (c) and (d). Target appearance changes dramatically through the whole video. As to a medium IoU threshold, for example the second and third matching strategy, their tracking performance are even worse.

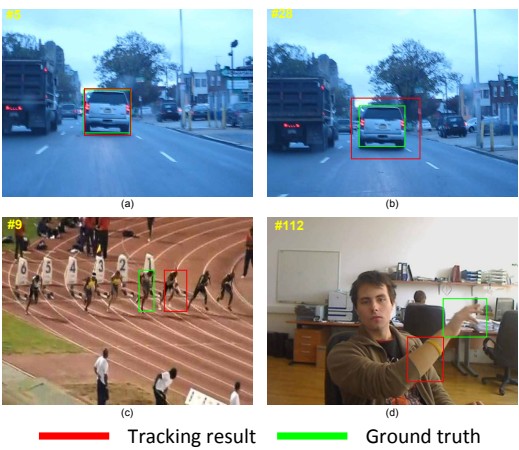

Figure 3: Qualitative evaluation of matching strategies. (a) is an early frame of first matching strategy, and (b) shows bounding box will get larger with time elapse. (c) and (d) show that trackers drift quickly to abrupt motion and deformation in fourth matching strategy.

**Analysis:** For the first matching strategy, due to back-propagation only goes through anchors with high IoUs, the tracker is able to follow up target object, but performs poor to provide tight bounding boxes. Eventually, the phenomenon in Figure 3 (b) would occur. In the fourth matching strategy, bounding box is accurate, but errors from classification would increase. That is owing to anchor points with low IoUs are equally treated with higher ones. Actually, higher IoUs should be more confident than lower ones. According to evaluation results of the second and third matching strategies, if choosing a medium IoU threshold, tracking performance would not be improved, instead get even worse.

We find the first matching strategy is good at classification, but poor in bounding box accuracy. Although loss function in the fourth matching strategy only contains a classification part, comparing with the first matching strategy, it performs better in providing tight bounding boxes rather than accurate classification.

### 3.5 TRACKING LOSS

In the paper, the key idea is to design a tracking loss to convert detection to tracking simultaneously considering the discrimination quality and bounding box accuracy. We propose to combine the first and fourth matching strategies together to define the new tracking loss. We believe this method could take advantage of pros and offset cons of each matching strategy. Tracking loss is defined as eq. (2). Where $a_i$ is the predicted probability of an anchor $i$ being an object in the first matching strategy. $a_i^*$ is the label of the ground truth. $p_i$ and $p_i^*$ are separately the predicted probability of and ground truth in the fourth matching strategy. $\alpha$ and $\beta$ are two variables to balance these two loss terms.

$$L_{TrackingLoss} = \alpha \sum_i L_{cls}(a_i, a_i^*) + \beta \sum_i L_{cls}(p_i, p_i^*) \qquad (2)$$

Tracking loss contains two matching strategies. Each has an independent RPN module. They share identical features from the backbone ZF network. With a Softmax Logistic Loss employed, we find

AO is significantly promoted to 0.52 in baseline experiment, and 0.46 in unsupervised experiment, as shown in Table 1. Here, the hyper-parameters $\alpha$ and $\beta$ are set to 1 and 10 respectively. Above evaluation results verify that tracking loss is effective to convert detection to tracking.

## 4 Network Compression

RPN is a relative large network to tracking. Additional actions should be taken to accelerate its speed. Moreover, shallow features are often field independent. A well-performed loss function should be robust to network variations. Each RPN has a backbone network for feature extraction. If backbone network is drastically compressed, could tracking loss still performs outstanding results? Learning from knowledge distillation, we propose a new network compression method to speed up, simultaneously to verify the robustness of tracking loss.

Hinton et al. (2015) proposed to distil knowledge from a large network into a small one. It was found that small networks trained from large networks could also offer similar performance than directly training a small network from scratch. Therefore, we propose to train the compressed ZF network from a pre-trained ZF network.

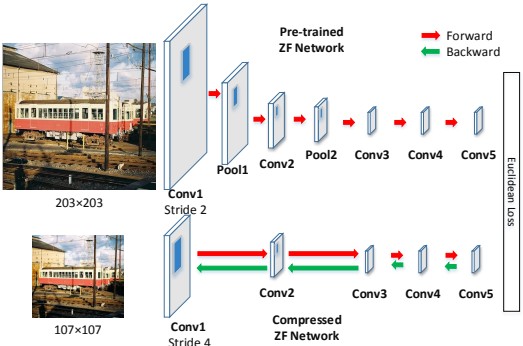

Figure 4: Training flow of compressed ZF network. We remove $Pool1$ & $Pool2$ in the compressed ZF network, and adjust the stride of $Conv1$ from 2 to 4. Other hyper-parameters remain invariant. We resize a batch of identical images in PASCAL VOC 2012 into two resolutions 203x203 and 107x107, feed them into two networks, and take $Conv5$ feature maps in the pre-trained ZF network as guidance to train the compressed ZF network with Euclidean Loss.

The training procedure of compressed ZF network is shown in Figure 4. To begin with, we use top layer feature maps of a pre-trained ZF network as ground truth whose weights won't be updated during training. The size of input images is 203x203. We believe it beneficial to make use of the knowledge learnt by pre-trained ZF network. Therefore, weights in compressed ZF network are initialized from the pre-trained ZF network. We take smaller images as input whose size is 107x107. In order to obtain an identical size of top layer feature maps, we adjust the stride of $Conv1$ in compressed ZF network from 2 to 4, and keep the stride of $Conv2$ unchanged. Moreover, we remove $Pool1$ and $Pool2$ layers. Those modifications will not change the size and amount of kernels in each layer. During training, we resize a minibatch of images into required resolutions. Then we separately feed those resized images into the pre-trained ZF network and the compressed ZF network. Euclidean loss is employed to guide training. Weights only update in the compressed ZF network during training. In addition, we remove the last pooling layer after $Conv5$, and normalize the feature maps with the mean and standard deviation of top layer feature maps of pre-trained ZF network on PASCAL VOC 2012 images (Everingham et al., 2015).

After training, we separately take pre-trained ZF and compressed ZF as backbone network of RPN and evaluate on VOT 2016 benchmark. Eventually, we get the same AO at 0.52 in baseline experiment and 0.46 in unsupervised experiment. Apparently, our network compression won't lose any old knowledge. That indicates tracking loss is robust to network compression and independent of feature extraction in backbone network. Surprisingly, the time of one forward pass with compressed ZF network decreases from 2.37s to 0.60s. The proposed network compression method significantly accelerates tracking speed four times.

## 5 ALGORITHM OVERALL

Previous evaluation results in Section 4 show that compressed ZF network could also provide high quality features like the non-compressed one. Therefore, we take compressed ZF network for feature extraction. Different types of loss functions approach Empirical Risk Minimization in different ways. We utilize four types of tracking loss to realize ensemble. Overall structure of our tracker is depicted in Figure 5.

On the left part of Figure 5, a compressed ZF network is used to extract features. On the right side, there are four independent branches with different types of loss function. Each branch contains two *Conv proposal* layer and two *Conv proposal cls score* layer separately for two parts in tracking loss whose weights would be updated during online training. Each loss branch also owns an independent scoring mechanism. Softmax Logistic Loss and Info-gain Loss take a *softmax* function. Sigmoid Cross-entropy Loss and Hinge Loss use a *sigmoid* function instead. The scores are used to identify whether candidate proposals belong to target object or background. Each baseline tracker only contains one single branch, corresponding to a dashed box in Figure 5. We average scores from four separate branches to define the similarity between a candidate proposal and ground truth. These four branches compose tracking loss ensemble.

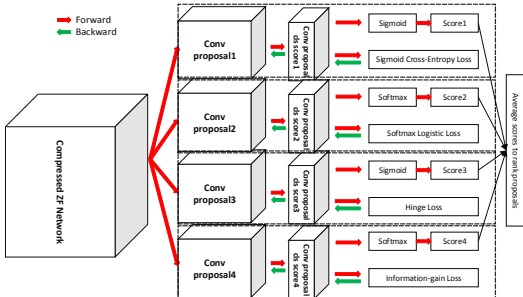

Figure 5: Overall architecture of our algorithm. A fine-tuned compressed ZF network is used for fast feature extraction. Four branches contain different types of tracking loss forming ensemble.

Weights in *Conv proposal layer* and *Conv proposal cls score layer* are randomly initialized before online tracking with a Gaussian distribution. The value $\alpha$:$\beta$ require to be determined before tracking. After fine-tuning, we use 1:10 in Softmax Logistic Loss and Information-gain Loss, 4:1 in Sigmoid Cross-entropy Loss and 3:9 Hinge Loss. The base learning rate is 0.0002 for Sigmoid Cross-entropy Loss, 0.0005 for other loss functions. Momentum is set to 0.9. Weight decay is 0.0005.

## 6 EXPERIMENT

Our tracker is implemented in MATLAB using Caffe deep learning framework. It is evaluated on a workstation with NVidia GeForce GTX Titan X GPU and Intel i7 3.6GHz CPU. The speed is about 1.6 FPS with a compressed ZF network and tracking loss ensemble. For a single loss tracker, the speed is 1.7 FPS. With respect to MDNet running at 1FPS, our tracker accelerates 60% in speed.

### 6.1 EVALUATION ON VOT 2016 BENCHMARK

We evaluate our tracker on VOT 2016 (Kristan et al., 2016) which contains a moderate scale dataset with 60 challenging video sequences. This benchmark pays more attention to quality of contents and annotations rather than quantity. A re-initialization protocol will be triggered whenever the predicted bounding box in any frame has zero overlap with the target ground truth. We use EAO, accuracy and robustness to judge tracking performance. AO measures the accuracy between predictions and ground truths. Robustness is defined as average failures of each tracking video sequence. EAO measures the expected no-reset overlap of a tracker running on a short-term sequence, which could balance the accuracy of successful frames and robustness of sequences with diverse lengths (Kristan et al., 2016).

We evaluate four baseline tracking loss and tracking loss ensemble. Each tracker takes compressed ZF network as backbone network. To make a horizontal comparison, we use 9 state-of-the-art algorithms on VOT 2016 Challenge (Kristan et al., 2016) as contrast, including CCOT (Danelljan et al., 2016), TCNN (Nam et al., 2016), SSAT, MLDF, Staple (Bertinetto et al., 2016), DDC, EBT (Zhu et al., 2016) SRBT and MDNet_N (MDNet without training on labeled videos). In Figure 6 and Table 2, Ours_softmax denotes the baseline tracker with single Softmax Logistic Loss, Ours_sigmoid stands for a tracker with Sigmoid Cross-entropy Loss, Ours_hinge for Hinge Loss, Ours_info1 for Information-gain Loss with matrix [0.9 0.1; 0.1 0.9], Ours_info2 for Information-gain Loss with matrix [0.8 0.2; 0.2 0.8], Ours_info3 for Information-gain Loss with matrix [0.7 0.3; 0.3 0.7]. Ours_ensemble denotes tracking loss ensemble, which is composed of four types of tracking loss, including Softmax Logistic Loss, Sigmoid Cross-entropy Loss, Hinge Loss and Information-gain Loss.

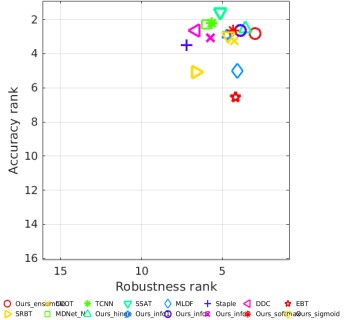

Figure 6: Ranking plot of Accuracy and Robustness. Right top trackers perform excellent and achieve a tradeoff between bounding box accuracy and discrimination quality.

Among all single tracking loss methods, Hinge Loss performs the best in EAO and robustness. Ours_info2 is the best among three Information-gain Loss based trackers. So we take $information$ $matrix$ [0.8 0.2; 0.2 0.8] in tracking loss ensemble. Ours_hinge and Ours_info2 exceed all contrast trackers in EAO and robustness. Softmax Logistic Loss and Sigmoid Cross-entropy Loss also exceed most contrast trackers. Figure 6 is the ranking plot of trackers in terms of accuracy and robustness. Tracking loss ensemble occupies the top right corner, achieves a tradeoff between accuracy and robustness. According to Table 2, tracking loss ensemble is the most robust one and achieves the first place in EAO, which outperforms state-of-the-art trackers, like CCOT, TCNN. MDNet_N falls behind due to its backbone network here is pre-trained on ImageNet rather than labeled videos.

Our proposed tracking loss algorithms (including Ours_hinge, Ours_info2 and Ours_ensemble) exceed all contrast trackers in EAO and robustness. But they fail to achieve good performance in accuracy. We suppose that might result from tracking loss only contains classification part, no bounding box regression part. Nevertheless, tracking loss is robust to keep up with target object without interruption. Eventually, overall performance is improved significantly and achieves the first in EAO which is the most significant indicator among the three. Actually, trackers are also ranked by EAO on VOT 2016 challenge (Kristan et al., 2016).

Table 2: EAO, Accuracy and Robustness comparing with trackers on VOT 2016 Challenge. $Red$ stands for ranking the first, $blue$ for the second and $green$ for the third.

| Trackers | EAO | Accuracy Rank | Robustness Rank |
|---|---|---|---|
| Ours_ensemble | 0.3498 | 2.80 | 2.98 |
| Ours_hinge | 0.3384 | 2.53 | 3.60 |
| Ours_info2 | 0.3365 | 2.63 | 3.88 |
| CCOT | 0.3310 | 3.23 | 4.28 |
| TCNN | 0.3249 | 2.22 | 5.67 |
| Ours_softmax | 0.3241 | 2.65 | 4.35 |
| Ours_sigmoid | 0.3210 | 2.97 | 4.65 |
| SSAT | 0.3207 | 1.55 | 5.15 |
| Ours_info1 | 0.3166 | 2.88 | 4.73 |
| MLDF | 0.3106 | 5.00 | 4.08 |
| Staple | 0.2952 | 3.52 | 7.23 |
| DDC | 0.2929 | 2.65 | 6.67 |
| Ours_info3 | 0.2928 | 3.07 | 5.73 |
| EBT | 0.2913 | 6.55 | 4.22 |
| SRBT | 0.2904 | 5.07 | 6.67 |
| MDNet_N | 0.2572 | 2.27 | 6.00 |

## 7 DISCUSSION

In this paper, we find that data augmentation will bring noise to train trackers. Pixels in different positions should be treated discriminatively. If sampling is subject to a Gaussian distribution, centric pixels are more likely to cover the target object than border ones. Due to translation invariant anchor box design of RPN, we could easily map anchor boxes to points in the top layer feature maps. Central feature points are highly confident to be positive. Therefore, we regard four central points as the ground truth. Confidence scores of different anchor points are defined by IoU. According to discrimination in confidence scores, we design four matching strategies and employ RPN in tracking. Finally, we combine two better strategies and propose the novel tracking loss. Such contraption realizes to treat different feature points discriminatively.

Moreover, we propose a network compression method and accelerate tracking speed four times without performance decline. That also implies tracking loss is robust to network variations.

Evaluation results on VOT 2016 show that two baseline tracking loss trackers and the tracking loss ensemble tracker outperform all state-of-the-art trackers in terms of EAO and robustness. That verifies tracking loss is effective to convert a popular detector RPN to a well-performed tracker. Treating feature points in different positions discriminatively could improve tacking performance indeed.

Feature engineering and feature aggregation would cost more time and computation during converting. Comparing with them, tracking loss is a much lighter method to convert rich features in detectors to trackers.

## 8 CONCLUSION

In the paper, we propose a novel tracking loss to convert an object detector to a well-performed robust tracker without extra time or computational consuming modifications (*e.g.* feature engineering and feature aggregation). On the basis of inaccuracy of sampling, tracking loss fully exploits the internal structure of top layer features of the detection network to treat feature points discriminatively. Such structure could provide high-quality discrimination and tight bounding boxes in tracking. Our network compression yields 4 times speedup. That also proves tracking loss is robust to network variations. We further employ tracking loss ensemble to promote the performance. Evaluation results on VOT 2016 show that two baseline tracking loss trackers and the tracking loss ensemble tracker outperform all state-of-the-art trackers in terms of EAO and robustness.

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
