# OpenReview forum: "Tracking Loss: Converting Object Detector to Robust Visual Tracker"
_ICLR.cc/2018/Conference — Reject_

### Official Review · AnonReviewer1 · 2017-11-26
**Technical achievements but limited novelty - perhaps more suited for a computer vision venue. Results not fully transparent.**

**Rating:** 5
**Confidence:** 4

**Review:**

This paper proposes a method to adapt a region proposal network (RPN) to the visual object tracking task. The authors also describe a method to compress the network to improve run-time performance. They also claim that an ensemble of the proposed trackers provides state-of-the-art performance on VOT 2016.

The main claimed contribution is "a novel tracking loss which successfully converts a pre-trained object detector RPN to a state-of-the-art visual tracker..." The idea to adapt an RPN to visual object tracking by generating samples for online learning was previously published in the CVPR 2016 workshops [1]. Some novelty is also claimed in the network compression, but it seems to be a straightforward implementation of knowledge distillation. In this reviewer's estimation, the novelty in this paper is limited to the specific design of the loss function described in Section 3.5.

The tracking loss is essentially a procedure to limit or gate back-propagation updates to proposal regions that have a high confidence to match the object being tracking. The intuition is that this strategy will enable better online learning and thus tracking performance. A small empirical study was conducted to determine which feature regions from the top layer of the RPN are most effective for this purpose. The authors argue that allowing only high matches could lead to centered but loose bounding boxes, while allowing further matches can improve the bounding box fit but might encourage drift. The loss function is combination of the two, with \alpha and \beta weighting the importance (it seems you only need one weight parameter here). No theoretical justification for the approach is given, it seems to be an ad hoc solution to adapt a region proposal architecture to perform online tracking.

The network compression in Section 4 seems to yield a nice increase in efficiency without any loss in performance. The network ensemble described in Section 5 improves tracking performance over a single network. These are nice technical improvements that push performance, but do not offer much in terms of novelty.

The proposed tracking network is tested on the VOT 2016 challenge data. The authors claim state-of-the-art performance on this dataset. The source code and raw results of participants of VOT 2016 are all publicly available - but unfortunately the no raw results or source code are provided for this paper either in supplementary material or in anonymous repository (it is not difficult to do this while keeping anonymity). Tracking papers usually provide some video results or, at minimum, still frames, to assist in the evaluation of performance, but none are provided here The Accuracy Rank and Robustness Rank numbers provided in Table 2 seem to be incorrectly computed - these numbers should be integers, see the VOT challenge report [2].

Pros:
+ Strong practical and technical improvements to push performance
+ State-of-the-art performance on VOT 2016

Cons:
- Difficult to verify results
- Limited novelty
- Numerous language problems, making the paper difficult to read and understand in many places.


[1] Zhu, G., Porikli, F., & Li, H. (2016). Robust visual tracking with deep convolutional neural network based object proposals on pets. In Proceedings of the IEEE Conference on Computer Vision and Pattern Recognition Workshops (pp. 26-33).
[2] VOT challenge report http://data.votchallenge.net/vot2016/presentations/vot_2016_paper.pdf

Minor notes:
* Clearly explain what an anchor point is
* Figures are introduced out of order
* What is the purpose of Figure 1? There seems to be no clear message
* "Although ground truths are not provided in tracking..." - what do you mean by this?
* Do you fix the aspect ratio of the anchors to 1:1 for tracking, or only to define the confidences in Fig 2?
* "RPN is a relative large network to tracking" - I understand what you mean but it is not written clearly
* Unclear how \alpha:\beta is computed
* Figure 6 is too small to read.
* Table 2 size should be increased.

---

### Official Review · AnonReviewer2 · 2017-11-27
**Mediocre paper without depth or significant contributions.**

**Rating:** 3
**Confidence:** 5

**Review:**

In this paper, the authors propose a novel tracking loss to convert the RPN to a tracker. The internal structure of top layer features of RPN is exploited to treat feature points discriminatively. In addition, the proposed compression network speeds up the tracking algorithm. The experimental results on the VOT2016 dataset demonstrate its efficiency in tracking.

This work is the combination of Faster R-CNN (Ren et al. PAMI 2015) and tracking-by-detection framework. The main contributions proposed in this paper are new tracking loss, network compression and results.

There are numerous concerns with this work:

1.	The new tracking loss shown in equation 2 is similar with the original Faster R-CNN loss shown in equation 1. The only difference is to replace the regression loss with a predefined mask selection loss, which is of little sense that the feature processing can be further fulfilled through one-layer CNN. The empirical operation shown in figure 2 seems arbitrary and lack of theoretical explanation. There is no insight of why doing so. Simply showing the numbers in table 1 does not imply the necessity, which ought to be put in the experiment sections.
2.	The network compression is engineering and lack insight as well. To remove part of the CNN and retrain is a common strategy in the CNN compression methods [a] [b]. There is a lack of discussion with the relationship with prior arts.
3.	The organization is not clear. Section 3.4 should be set in the experiments and Section 3.5 should be set at the beginning of the algorithm. The description of the network compression is not clear enough, especially the training details.  Meanwhile, the presentation is hard to follow. There is no clear expression of how the tracker performs in practice.
4.	In addition, VOT 2016, the method should evaluate on the OTB dataset with the following trackers [c] [d].
5.	The evaluation is not fair. In Sec 6, the authors indicate that MDNet runs at 1FPS while the proposed tracker runs at 1.6FPS. However, MDNet is based on Matlab and the proposed tracker is based on C++ (i.e., Caffe).

Reference:
[a] On Compressing Deep Models by Low Rank and Sparse Decomposition. Yu et al. CVPR 2017.
[b] Designing Energy-Efficient Convolutional Neural Network Using Energy-Aware Pruning. Yang et al. CVPR 2017.
[c] ECO: Efficient Convolution Operators for Tracking. Danelljan et al. CVPR 2017.
[d] Multi-Task Correlation Particle Filter For Robust Object Tracking. Zhang et al. CVPR 2017.

---

### Official Review · AnonReviewer3 · 2017-11-29

**Rating:** 4
**Confidence:** 4

**Review:**

The paper derives from the popular RPN method and hypothesizes that pixels at different positions, in top feature maps for tracking, should be treated with various emphasis; and they design four matching (anchors and ground truth boxes) strategies to explore which of them are proper for tracking. This is most important idea proposed in the paper. The model compression part is for speed consideration and the model ensemble idea is a general trick to further improve the performance on VOT 2016.

However, I think the paper is not ready for ICLR yet due to several reasons.

- Novelty. The newly proposed tracking loss is still the standard, well-known classification loss with the novel part that "matching strategy" is different. I do like the analysis part as to why the first and fourth strategy is chosen in section 3.4. But this seems not to be a big difference from previous work.

- Experiments not enough and not organized well. The paper spares the second half of its novelty to network compression (of which there is barely new; but I do understand the necessity to speed up algorithms in real-time products); however, there is no clear comparison (table or figure) to point out how the acceleration is. Only some words in the paper: "The proposed network compression xxx four times", "our tracker accelerates 60% in speed", etc. Figure 6 is not depicted in a clear manner.

Table 2 lists the performance comparison across methods; what is the meaning of Information-gain loss? It is fair to compare the model ensemble version of yours to other methods? How about the results of the ensemble version of other methods? There is no ablative study in the experiment also.

The paper has many presentation drawbacks (syntax errors, format issues, etc.) For example, "Basing on knowledge distillation theory xxx" -> Based. I won't list all that I find here.

---

### Decision · Program_Chairs · 2018-01-29
**ICLR 2018 Conference Acceptance Decision**

**Decision:**

Reject

**Comment:**

While the reviewers are interested in the work, they find it not good enough for ICLR.
There are a number of limitations, including experimental results and presentation / language
There is no author  response / revision.